# Ablation Behavior of Silicone Rubber-Benzoxazine-Based Composites for Ultra-High Temperature Applications

**DOI:** 10.3390/polym11111844

**Published:** 2019-11-08

**Authors:** Jinglong Gao, Zhixuan Li, Jiayi Li, Yanhui Liu

**Affiliations:** 1School of Material Science and Engineering, Shenyang Ligong University, Shenyang 110159, China; jlgao@sylu.edu.cn (J.G.); a13664119712@163.com (Z.L.); 2School of Metallurgy, Northeastern University, Shenyang 110004, China; xflgjl@163.com

**Keywords:** benzoxazine resins, ablation composites, ultra-high temperature, oxygen enrichment

## Abstract

A novel type of silicon rubber composite with benzoxazine resins (BZs) and ZrO_2_ was prepared. The ablative response of the composites was investigated. The results showed that the composites with BZs had superior thermal stability and higher resides compared to the pristine composites. The linear ablation rate of the composites decreased significantly with the increase in ZrO_2_ content. The maximum back-face temperature of the burnt samples was no more than 100 °C for the obtained composites. Three major ablation processes were carried out simultaneously during the ablation processing. These mainly involved the carbonization of the composite, and the formation of ceramic compounds such as SiC and ZrC, as well as the shielding effect of the ablated layer, which subsequently enhanced the ablation resistance of the composites.

## 1. Introduction

With the development of hypersonic aerospace vehicles, high-energy propellants have found widespread applications. However, due to the increase in thermal transmission and the thinning of the shell, the thermal shielding problem becomes more prominent. Polymer matrix-based materials play a crucial role in the thermal protection systems which provide thermal insulation to solid rocket motors by the ablation process [1,2,3]. The thermal shielding effect depends on complicated physical–chemical changes which involve melting, sublimation, degradation, ceramic reaction and carbonization during the ablation process. Thermal insulation layer materials can absorb heat by two main methods, a physical process including melting and sublimation, as well as a chemical process which involves pyrolysis, ceramic reaction, and carbonization. Through the above processes, it forms a carbonization layer or ceramic layer, which subsequently helps to achieve thermal protection. It has been established that the formation of the refractory char on the surface of the material is very resistant to peel off in a hyperthermal environment characterized by high-pressure eroding gas in a solid rocket motor [4,5].

As a kind of heterochain elastomer, silicone rubber (SR) has been used as the substrate material for ablative composites because of its excellent properties, such as high temperature resistance, pyro-oxidation resistance and shaping [6,7,8,9,10,11]. However, the intrinsic shortcomings of SR, i.e., low mechanical strength and poor carbon residue rate, have restricted its applications [3,12]. In order to overcome such a problem, SR is usually blended with high-melting inorganic fillers, such as inorganic powders ZrO_2_, ZrC and ZrB_2_ [6,7]. Kim et al. investigated the effect of the incorporation of carbon fibers (CFs) and silicon carbide powders (SCPs) on the ablation properties of SR. The results showed that an efficient improvement of the ablation property was observed [12]. The Yang group studied the mechanical properties, thermal stability and ablation behaviors of SR composites incorporated with ZrC or ZrO_2_. They concluded that with the increase in ZrO_2_ or ZrC content, the linear and mass ablation rates of the composites gradually decreased [6].

However, this also established that adding high-melting inorganic fillers makes the surface layer brittle, with a low carbon content and high density. To deal with the carbon residue problem, it is effective to add a polymer resin which is of high carbon content, into the silicone rubber–inorganic filler composites. Benzoxazine resins (BZs) are a type of highly cross-linked aromatic polymer with outstanding advantages, including excellent thermal resistance, good flame retardancy, and near-zero volumetric shrinkage upon curing and high carbon residue [13,14,15,16]. These extraordinary properties make BZs the ideal resin to be used in SR composites for ultra-high temperature applications.

In this study, SR composites containing BZs and ZrO_2_ were prepared by melt mixing and co-curing. The thermal stability and ablation behaviors of the composites were investigated by thermal analysis and oxyacetylene torch. The ablation mechanism is discussed in detail.

## 2. Experimental Section

### 2.1. Materials

Polydimethylsiloxane (PDMS) and polymethylphenylsiloxane (PMPS) were purchased from the Shanghai Resin Co., Shanghai, China. The number average molecular weight and the vinyl contents of PDMS were 550,000 and 0.15 mol %, respectively. The number average molecular weight and the pehnyl contents of PMPS were 600,000 and 11.3 mol %, respectively. The BZs were produced by the ACO Pharm Co., Ltd., Shanghai, China. Fumed silica (Shenyang chemical Co., Ltd., Shenyang, China) with a specific surface area(BET) of 250 m^2^/g was used as a reinforcing agent to enhance the mechanical strength of the composites. Dicumyl peroxide (DCP) was used as a curing agent, which was supplied by Shanghai Shanpu chemical Co., Ltd., Shanghai, China. The zirconia with an average particle size of 60 nm was purchased from Shanghai Naiounano Technology Co., Ltd., Shanghai, China. The chopped Toray’s T300 carbon fibers (6 mm) were produced by the Nanjing Man Kate Technology Co., Ltd., Nanjing, China. Prior to the preparation of the composites, carbon fiber was calcined in a furnace at 400 °C for 2 h to eliminate adsorbed species. All materials were used as received without further purification except the carbon fibers.

### 2.2. Sample Preparation

Formulations of the silicone rubber–benzoxazine-based composites are shown in Table 1. The composites were prepared using a two-roll mill at 85 °C at the speed of 40 rpm. Silicone rubber was added first and followed with fumed SiO_2_. In order to get the homogenous sample, carbon fibers and BZs were continuously added to silicone rubber. After melt blending, ZrO_2_ and DCP were added into the system. The whole process lasted 30 min. The refolded sheet was cut into a cylinder with a thickness of 10 mm and a diameter of 30 mm. The cylinder was then molded and vulcanized by compression molding at 175 °C and 10 MPa for 30 min. The secondary vulcanization process was carried out at 220 °C for 2 h.

### 2.3. Characterization and Ablation Experiment

#### 2.3.1. Ablation Experiment

The ablation experiments were conducted using an oxyacetylene torch according to GJB 323A-96. The flow rates of oxygen and acetylene were 0.30 and 0.24 m^3^/h, respectively. The specimen was placed vertically to the flame direction. The temperature profiles were acquired at 10 mm under the hot surface of the composites. The K-type thermocouple was arranged in a blind hole drilled at the bases of the cylindrical shaped hold instruments. The distance between the nozzle tip of the oxyacetylene gun and the front surface of the specimen was 10 mm. The inner diameter of the nozzle tip was 2.0 mm. The ablation test time for each sample was 20 s. The line ablation rate (*R*_d_) can be calculated according to Equation (1):(1)$$R_{d}=\frac{\Delta d}{t}=\frac{d_{1}-d_{2}}{t}$$
where *d*_1_ and *d*_2_ are the thickness of the specimen before and after ablation testing; *t* is the ablation time (20 s).

#### 2.3.2. Sample Characterization

The microstructures of the composites after ablation were characterized by scanning electron microscopy and energy depressive spectrometry (SEM–EDS, S-3400N, Hitachi High-Tech Manufacturing & Service Co., Ibaraki, Japan).

The thermogravimetric analysis (TGA) (Netzsch STA 449F3, Munich, Germany) was used at a heating rate of 10 °C/min from room temperature to 800 °C under static air atmosphere.

The crystalline phase of samples was identified by XRD (MPDDY2094, Panalytical, Almelo, Netherlands), with Cu Kα radiation (λ = 1.54056 Å) at the scanning rate of 6 °/min from 5° to 90° and subsequently the peaks were identified by the X’ Pert High Score Plus software (VX6000R, Jinhu Juke Instrument Co., Ltd., Huai’anHuai’an, China).

## 3. Results and Discussion

### 3.1. TGA

The TGA curves of samples BZ-0 and BZ-1 are shown in Figure 1. It is clear that sample BZ-1 demonstrates better thermal stability and less weight loss than sample BZ-0. The rapid degradation of sample BZ-0 occurred at 400 °C and the weight loss was approximately 50% at 490.2 °C, whilst BZ-1 began to decompose at 450 °C and the weight loss was about 50% at 523.3 °C. Moreover, there is also an additional decomposition step from about 200 to 450 °C, with a weight loss of about 10%. Samples BZ-0 and BZ-1 exhibit a total weight loss of 84.21 and 79.36%, respectively. This suggests that benzoxazine with high carbon content was likely to increase carbon residue and form an ablation layer at high temperature, which subsequently leads to the excellent thermal resistance of sample BZ-1.

### 3.2. Ablation Properties of the Composites

Figure 2 shows the effect of the concentration of BZs on the linear ablation rate of the composites. With the increase in BZ content, the linear ablation rate of the composites firstly decreases and then increases. When the content of BZs increases from 0 to 20 phr, the linear ablation rate is reduced by 44.92%. Meanwhile, it has been established that with the addition of BZs, the linear ablation rate values are all below 0.08 mm s^−1^. Compared to those of the commonly used silicone rubber-based composite from the literature (0.083 ~ 0.108 mm s^−1^) [17], it is concluded that the ablation resistance of the composites can be effectively enhanced by the addition of BZs.

The impact of ZrO_2_ content on the linear ablation rate of the composites is shown in Figure 2. The experimental results show that the ablation resistance of the composites is improved by incorporation of ZrO_2_, which has a high melting point. With the increase in ZrO_2_ content, the linear ablation rate of the composites gradually decreases. The linear ablation rate of the specimen decreases by 60.48% with the addition of 20 phr BZs and 20 phr ZrO_2_ compared to that of the neat specimen.

### 3.3. Burnt Structure and Composition Analysis of the Composites

The image of sample BZ-2 after ablation is shown in Figure 3a. Under exposure to the Oxyacetylene flame, fumed silica, which has a lower melting point (about 1650 °C), melted easily into the less viscous spheres which were rapidly flowing away from the zone touched by the flame plume to the external edge of the flame. The spheres were easily removed, leaving the charred surface unprotected due to the poor compatibility between the melt and the substrate. A high erosion rate was experienced by the material. An ablated layer with cracks was easily peeled off the matrix layer.

The image of the Zr-2 sample is shown in Figure 3b. The whole surface of the ablated material is relatively homogeneous and smooth, compared to that of sample BZ-2. Furthermore, no obviously melted spheres can be observed even from the edge of the sample. Under exposure to the oxyacetylene flame, fumed silica and ZrO_2_ melted together to form blend droplets. The viscosity of these melting droplets increased in the presence of ZrO_2_, which is of higher melting point (about 2700 °C). The movement of the droplets from flame center region towards the edge of the flame was therefore hindered. In fact, this quantity of ZrO_2_ was able to effectively freeze the droplets under the zone hit by the flame and confirm the enhanced stickiness and integrity of the char. Combined with analysis of the SEM, it is clear that the burnt substrate increased its binding capability and compatibility with the melt. The erosion rate experienced by the Zr-2 decreased accordingly.

SEM images of samples BZ-2 and Zr-2 after ablation are shown in Figure 4. As illustrated in Figure 4a, the surface of the char layer of sample BZ-2 is rough and covered by particles with sizes ranging from several hundreds of nanometers to less than 2 μm. Many deep holes with different diameters (up to 40 μm) are obviously observed on the ablated longitudinal profile of sample BZ-2, as shown in Figure 3b. Considering Figure 4c, uniformly distributed spherical particles with diameters of less than 0.5 μm can be observed. The ablation layer surface of sample Zr-2 is relatively flat and dense, compared to that of sample BZ-2. Analyzing the longitudinal profile of the Zr-2 specimen, as demonstrated in Figure 4d, the pores are both smaller and fewer in number than those of sample BZ-2. As the heat protection shield, the formation of the more dense and rigid refractory layer is very resistant to not only the convective heat transfer of the surface, but also erosion by surface gas flow during the ablation process [5].

The elemental composition of the ablation surface layers was analyzed by EDS, as shown in Table 2. Compared to that of the BZ-2, there is higher Si, C and Zr content, with lower O content in the surface layer for Zr-2.

Figure 5. reports the results of the XRD characterization carried out on the burnt specimens. Figure 5a shows the XRD patterns of the Zr-2 ablation surface layer. Compared to those of the BZ-2, two additional types of diffraction peaks are observed, which can be assigned to the ZrO_2_ and ZrC. As shown in Figure 5b, for the ablation surface of BZ-2, two diffraction peaks can be observed, which correspond to the reflection of SiO_2_ and SiC, respectively.

During oxy-acetylene tests, the composites underwent a series of complex chemical reactions and phase transitions. The proposed reactions during the ablation process are shown as Equations (2)–(5). The ablation caused the chemical degradation of the polymer matrix, leading to the conversion of organic carbon to inorganic carbon. In the presence of SiO_2_ and ZrO_2_, carbothermal reductions producing SiC and ZrC took place. As flame retardants, the formation of ceramic compounds, such as SiC and ZrC, are beneficial to the anti-ablation properties of the composites.
(2)$$\mathrm{Silicon}\;\mathrm{rubber}\;\mathrm{matrix}\;+\;\mathrm{BZs}\to {\;C\;+\;\mathrm{SiO}}_{2}\;+\;\mathrm{Si}$$
(3)$${3C\;+\;\mathrm{SiO}}_{2}\;=\;2\mathrm{CO}\;+\;\mathrm{SiC}$$
(4)$${2\mathrm{SiO}}_{2}\;+\;5C=\mathrm{SiC}\;+\;4\mathrm{CO}\;+\;\mathrm{Si}\;$$
(5)$${3C\;+\;\mathrm{ZrO}}_{2}=\mathrm{ZrC}\;+\;2\mathrm{CO}$$

The phase transitions were carried out simultaneously during the ablation processing. The SiO_2_, SiC, and ZrO_2_ phases had undergone melting and/or sublimation under a condition of ultra-high temperature (more than 2700 °C). Those inorganic components showed excellent ablation resistance which involves endothermic behavior, as shown in Equations (6)–(10):(6)$${\mathrm{SiO}}_{2}\;(s)\;\to {\mathrm{SiO}}_{2}\;(l)$$
(7)$${\mathrm{SiO}}_{2}\;(l)\;\to {\mathrm{SiO}}_{2}\;(g)\;$$
(8)$${\mathrm{SiO}}_{2}\;(s)\;\to {\mathrm{SiO}}_{2}\;(g)$$
(9)SiC (s) →SiC (g)
(10)$${\mathrm{ZrO}}_{2}\;(s)\;\to {\mathrm{ZrO}}_{2}\;(l)$$

Therefore, this suggested that the addition of BZs helps to produce more charcoal during the pyrolysis, which provides myriad of reactants for carbonation reaction. The addition of ZrO_2_ could facilitate the cross-linking reaction of ZrC under high temperature to form a much more compact cross-linking structure.

### 3.4. The Back Surface Temperatures

Figure 6 shows the back-face temperature–time relation curve of the ablative composites recorded at 10 mm under the surface of the tested samples during the ablation testing. It shows that the maximum back-face temperatures of the BZ-2 and Zr-2 are 89 °C and 94 °C, respectively. The temperature enhancement was rapidly increased in the 50 s ablation test for Zr-2 in comparison with that of BZ-2. This result can be directly related to the fact that increasing the amount of inorganic filler (ZrO_2_) reduces the related level of the polymer matrix. The presence of a higher fraction of inorganic component increased the thermal conductivity of the composites.

## 4. Conclusions

The obtained TGA results showed that the composite with BZs had superior thermal stability and higher resides than the pristine composite. With the increase in the BZ content, the linear ablation rate of the composites firstly decreased and then increased. The linear ablation rate of the composites obviously decreased in the presence of ZrO_2_, even with less content. In the case of the BZ-2 sample, as the matrix charred, the pristine resin matrix was not able to effectively hold the burned substrate, leading to a high erosion rate. In the case of the Zr-2 sample loaded with ZrO_2_, the complex physical–chemical reaction processes were carried out simultaneously during the ablation processing. The fumed silica melted together with ZrO_2_, thereby producing a viscous layer. The movement of the droplets from the flame ablation region towards the edge of the flame was hindered. The integrity and compatibility of the ablation surface with the melt also increased. The XRD results also confirmed that the formation of ceramic compounds such as ZrO_2_, ZrC and SiC prevent further decomposition of the composites, which enhances the ablation resistance of the silicone rubber composites. The temperature peak of thermocouple is 89 °C for the BZ-2 compared with that of 94 °C for Zr-2.

## Figures and Tables

**Figure 1 polymers-11-01844-f001:**
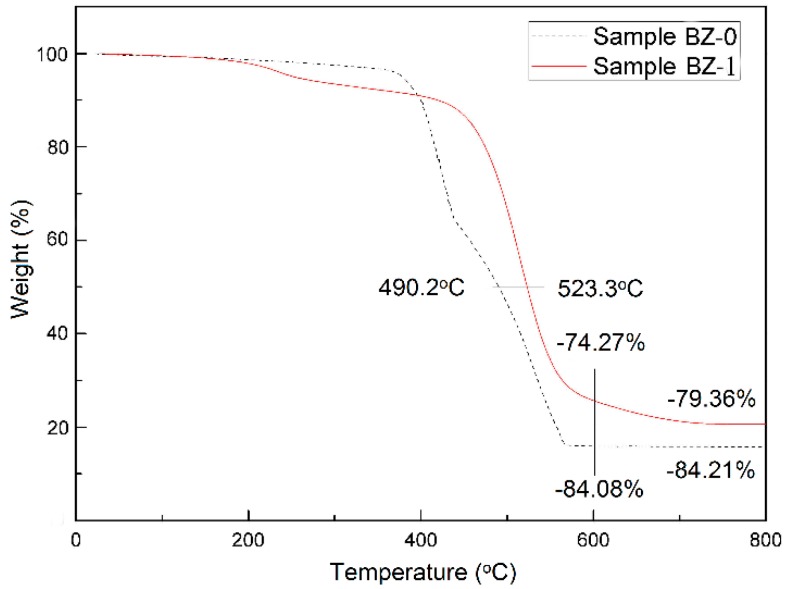
TGA curves of samples BZ-0 and BZ-1.

**Figure 2 polymers-11-01844-f002:**
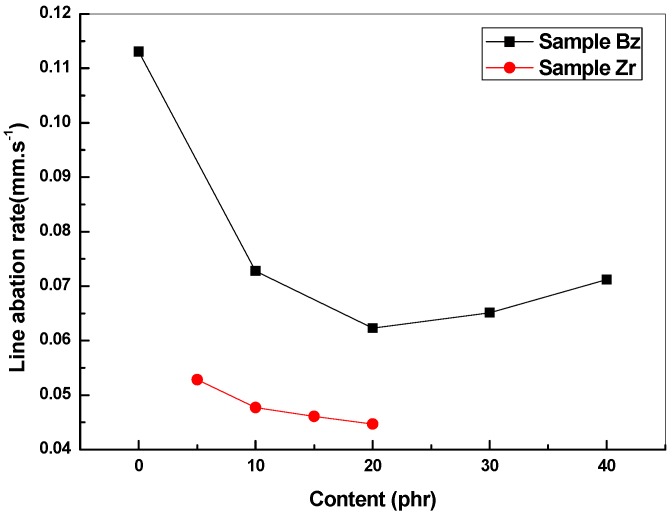
Linear ablation rate of the composites with different contents of BZs and ZrO_2_.

**Figure 3 polymers-11-01844-f003:**
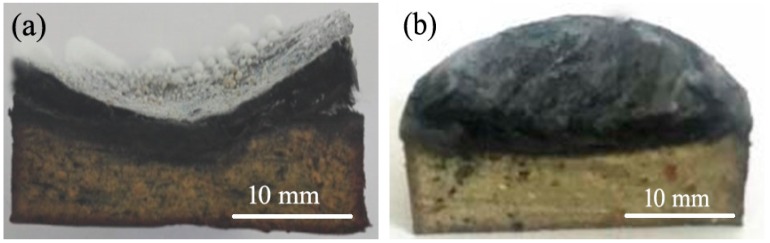
Post-burnt images of the composites. (**a**) BZ-2 and (**b**) Zr-2.

**Figure 4 polymers-11-01844-f004:**
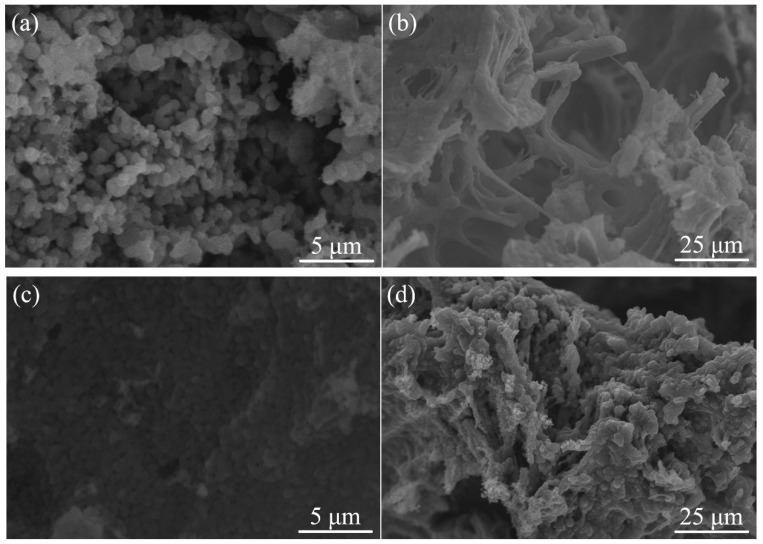
The surface and longitudinal profile morphology of the BZ-2 specimen (**a**,**b**) and Zr-2 specimen (**c**,**d**) after ablation.

**Figure 5 polymers-11-01844-f005:**
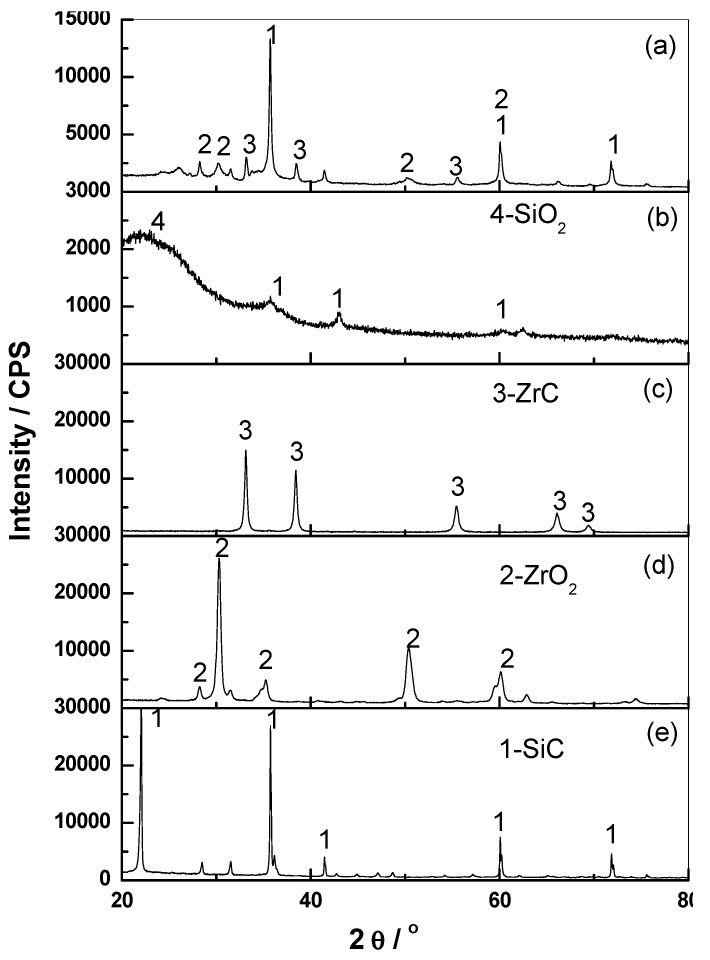
XRD patterns of the specimens after ablation, (**a**) Zr-2, (**b**)BZ-2, (**c**) ZrC, (**d**) ZrO2, and (**e**) SiC.

**Figure 6 polymers-11-01844-f006:**
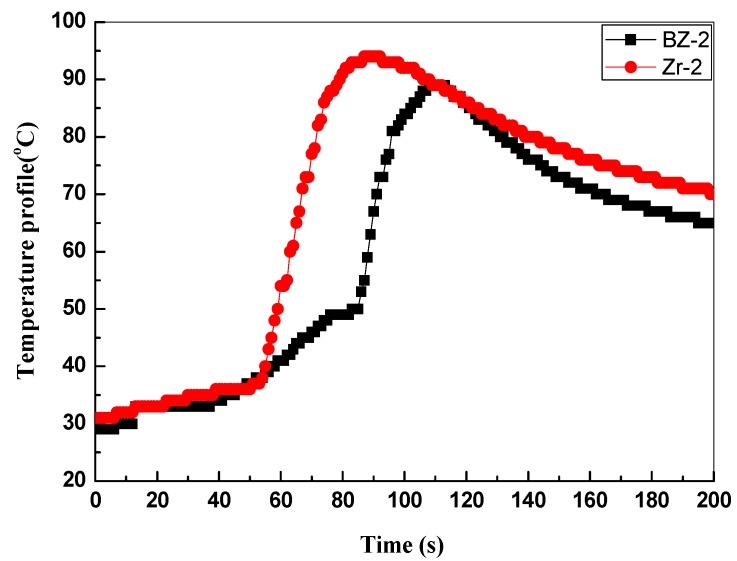
Back-face temperature curves of Zr-2 and BZ-2 specimens during the ablation testing.

**Table 1 polymers-11-01844-t001:** Formulations of the silicone rubber–benzoxazine-based composites.

Sample	PDMS (phr)	PMPS (phr)	SiO_2_ (phr)	BZs (phr)	CF (phr)	ZrO_2_ (phr)
BZ-0	50	50	30	0	10	0
BZ-1	50	50	30	10	10	0
BZ-2	50	50	30	20	10	0
BZ-3	50	50	30	30	10	0
BZ-4	50	50	30	40	10	0
Zr-1	50	50	30	20	10	5
Zr-2	50	50	30	20	10	10
Zr-3	50	50	30	20	10	15
Zr-4	50	50	30	20	10	20

**Table 2 polymers-11-01844-t002:** EDS analysis of the BZ-2 and Zr-2 specimens after ablation.

Specimens	O (%)	Si (%)	Zr (%)	C (%)
BZ-2	54.11	43.66	-	2.23
Zr-2	42.91	46.25	6.87	3.97

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
