# Peer review of "Ablation Behavior of Silicone Rubber-Benzoxazine-Based Composites for Ultra-High Temperature Applications"

_polymers, 2019, doi:10.3390/polym11111844_

Round 1

Reviewer 1 Report

Jinglong Gao et al., reported a study on “Ablation behavior of silicone rubber/benzoxazine based composites for ultrahigh temperature applications”. The paper is worth publishing in Polymers, after major revisions.

From the reference [5] it is demonstrated that ZrO2 decreases the linear and the mass ablation rate of the composite, so it is not clear what advantage there is in preparing composites with BZ also containing ZrO2.

As for the analyzes on the prepared samples, the authors show the linear ablation rate of all the samples (which confirms that ZrO2 does not have an improving effect), but they show TGA data only for BZ-0 and BZ-1, and SEM, EDS, XRD only for the ZR-2 sample, without explaining why they chose to show only the results related to this particular sample, whose line ablation rate is the worst.

The data reported are clearly insufficient and the authors have to show those of the other samples as well. This reviewer suggests reporting TGA, SEM, EDS, XRD data of ZR-2 and BZ-2 in the paper and to insert the results of all the other samples in Supplementary Materials, in order to allow a comparison of the behavior of all the samples.

The conclusions and the bibliography are very poor and need to be improved.

In conclusion, this referee believes that the subject of this paper is appropriate for Polymers and that it is of some interest, but for a correct evaluation of the behavior with the different BZ content, all the data have to be provided.

This referee suggests a revision of the English language, some light language mistakes have to be corrected.

Reviewer 2 Report

Include manuscripy changes in different color, maybe red

a) Manuscript has many typos, revise carefully and correct it b) Manuscript needs revision by English native c) Line 58, include information about purification compounds or used as received d) Line 105 change “200 oC to 450 °C” for 200 to 450 °C e) TGA results, include TGA (10 wt% loss, °C) and Char yield at 800 °C f) Change “84.21% and 74.36%” for 84.21 and 74.36% g) Improve quality of Figure 3 at minimal 300 dpi h) Figure 4, why Intensity in a.u., al figure a, b, c, d, and d have scale from 0 to 30,000 i) Figure 5 looks extremely poor, it needs to apply for FT-IR tools due to smoot, and normalize, but special SMOOT. Improve this figure or remove j) Figure 5, change Transmittance (%) for Transmittance (a.u.) k) Improve conclusion and include important results l) Manuscript needs more characterization techniques m) manuscript has some interesting results but doesn´t have discussion n) Revise some paper and think it is correct to start discussion with TGA

Round 2

Reviewer 1 Report

The article can now be published.

Reviewer 2 Report

Manuscript is accept in present form